# A Systematic Review of Circulating Tumor Cells Clinical Application in Prostate Cancer Diagnosis

**DOI:** 10.3390/cancers14153802

**Published:** 2022-08-04

**Authors:** Dmitry Enikeev, Andrey Morozov, Diana Babaevskaya, Andrey Bazarkin, Bernard Malavaud

**Affiliations:** 1Department of Urology, Medical University of Vienna, 1090 Vienna, Austria; 2Institute for Urology and Reproductive Health, Sechenov University, 119991 Moscow, Russia; 3Institute for Clinical Medicine, Sechenov University, 119991 Moscow, Russia; 4Department of Urology, Institut Universitaire du Cancer, 31059 Toulouse, France

**Keywords:** systematic review, prostate cancer, biomarker, diagnostic, circulating tumor cells (CTC)

## Abstract

**Simple Summary:**

Cell-dependent and cell-independent information drawn from the blood stream were merged into the attractive term “liquid biopsy” and tentatively applied to most segments of cancer management: detection, risk-stratification, personalization of care and follow-up. However, the robust science behind liquid biopsies has not been widely used, thereby remaining a latent and possibly undervalued instrument. Here, we conducted a systematic review of CTCs in prostate cancer management to summarize their use in clinical practice.

**Abstract:**

The purpose of the review is to summarize the recent data on circulating tumor cells (CTC) use in clinical practice. We performed a systematic literature search using two databases (Medline and Scopus) over the past five years and the following terms: (CTC OR “circulating tumor cells” OR “liquid biopsy”) AND prostate. The primary outcome was CTC predictive value for prostate cancer (PC) progression and survival. The secondary outcomes were the CTC predictive value for therapy response and the results of CTC detection depending on the assessment method. In metastatic PC, the CTC count showed itself to be a prognostic marker in terms of clinically important features, namely survival rates and response to treatment. CTC concentration was significantly associated with the overall survival and progression-free survival rates. A strong association between the overall survival or progression-free survival rate and CTC concentration could be observed. Variant-7 androgen receptors-positive (AR-V7-positive) patients showed a poor response to androgen receptor signaling (ARS) inhibitors, but this did not compromise their response to taxanes. In localized PC, only positive Cluster of Differentiantion 82 protein (CD82+) correlated with a higher survival rate. CTC count and AR-V7 expression showed itself to be a valuable biomarker for survival in metastatic PC and response to ARS-inhibitors. CTC diagnostic performance for localized PC or for screening and early detection is not high enough to show additional value over the other biomarkers.

## 1. Introduction

The natural history of solid tumors is driven by cancer heterogeneity, both spatial within the various sites of the metastatic disease and temporal at the different phases of tumor evolution [1].

After world war 2 (WW2), the pivotal work of Zeidman on the mechanisms of tumor emboli and metastasis unveiled the role of circulating tumor cells (CTCs) in the dynamics of cancer dissemination [2,3]. It is only recently that this paradigm has been fully validated by advances in the technology of capture and characterization of cancer cells and aggregates from blood samples in most solid cancer types, including prostate cancer (PC). At the same time, molecular techniques revealed that the blood compartment was also rich in cell-independent actors such as circulating tumor deoxyribonucleic acid (DNA) [4], cell-free micro-ribonucleic acid (micro-RNA) and extracellular vesicles [5].

Cell-dependent and cell-independent information drawn from the blood stream were merged into the attractive term “liquid biopsy” and tentatively applied to most segments of cancer management: detection, risk-stratification, personalization of care and follow-up.

However, as exemplified in a recent review for prostate cancer [6], the robust science behind liquid biopsies has failed to be used on a routine basis, thereby remaining a latent and possibly undervalued instrument. Here, we conducted a systematic review of CTCs in prostate cancer management according to the Preferred Reporting Items for Systematic Reviews and Meta-Analyses (PRISMA) guidelines.

## 2. Materials and Methods

### 2.1. Search Strategy, Inclusion Criteria

The detailed search strategy and review protocol have been published in Prospero (CRD42021239981). The present review followed the PICOS process (Patient, Intervention, Comparison, Outcomes, Studies):

P—patients with prostate cancer

I—detection of CTC in blood

C—histology

O—diagnostic accuracy, predictive value of CTC for cancer progression, survival, treatment response

S—all kinds of original studies

A systematic literature search was performed by scanning the Medline (Pub-Med) and Scopus databases over the past five years using the following search terms: (CTC OR “circulating tumor cells” OR “liquid biopsy”) AND prostate. Two authors (AM and AB) independently reviewed headings and abstracts to exclude irrelevant publications such as reviews, comments, papers in languages other than English and articles that dealt with other PC biomarkers or with conditions other than prostate adenocarcinoma (BPH, rare prostatic malignancies, etc.). In the event of disagreement between the reviewers, articles were retained for the following stage in the selection process.

After in extenso review of the publication, two readers (AM and AB) excluded those exclusively focused on laboratory techniques with no clinical relevance. In the event of disagreement, AM and AB sought to justify their decision and tried to resolve the disagreement. If they failed to reach an agreement, a senior researcher (DE) made the final decision. The present systematic review ultimately covered all original articles that addressed the clinical relevance of CTC in PC diagnostics and prognosis for the past five years.

### 2.2. Data Extraction Outcomes

Raw data, such as number of treated patients, cancer stage and treatment, methodology of CTC and markers expressed on CTC measuring, diagnostic performance was extracted manually from the articles.

The primary outcome was the CTC predictive value for PC progression and survival.

Secondary outcomes were the CTC predictive value for treatment response and results of CTC detection depending on the assessment method.

Due to the high heterogeneity in the studies with regard to methodology and the absence of a control group in most investigations, it was not possible to perform a meta-analysis, although a qualitative narrative synthesis was produced from the published literature.

### 2.3. Studies Quality Assessment

The level of evidence (LE) for each study was estimated according to the Oxford Centre for Evidence-based Medicine scale [7].

## 3. Results

### 3.1. General Characteristics of the Sample

After applying all the selection criteria, the final sample comprised 46 articles on metastatic PC [8,9,10,11,12,13,14,15,16,17,18,19,20,21,22,23,24,25,26,27,28,29,30,31,32,33,34,35,36,37,38,39,40,41,42,43,44,45,46,47,48,49,50,51,52,53,54] (Appendix A) and 15 articles on localized PC [55,56,57,58,59,60,61,62,63,64,65,66,67,68,69] (Appendix A). Three articles focused on CTC application in PC screening and early diagnosis [70,71,72] (Appendix A) (PRISMA statement, Figure 1).

### 3.2. Methods of CTC Capture and Characterization

Analyzing this field of diagnostic biology is hampered by the high heterogeneity observed in the technology designed to capture CTCs, the methods to characterize cells of epithelial lineage, the expression of results and the correlations researched with various clinical outcomes.

CellSearch^®^ was compared to CellCollectorTM and EPISPOT assay in 86 high-risk localized PC (LE 2b) [56]. Tests were repeated three months after prostatectomy in 52 patients, and the concordance was evaluated on the combined collection. A positive result was defined as the detection of at least one CTC. Although all three methods were positive in approximately 55% of the samples, they were rarely in agreement. CellSearch^®^ concurred with CellCollectorTM and EPISPOT in 56% and 59.9%, respectively, while the three techniques were concordant only in a minority (37.4%).

The same three techniques were tested in a cohort of 104 newly diagnosed high-risk prostate cancer patients, 19 of whom later proved metastatic (LE 2c) [65]. No tests correlated with age or prostate-specific antigen (PSA) and only CellSearch^®^ detected CTCs significantly more often in metastatic than in localized disease patients (61.1% vs. 14.3%, *p* < 0.001). As a single predictor, it was predictive (AUC ROC: 0.76) of the metastatic stage to an extent comparable to the classical D’Amico’s risk factors (PSA, Gleason sum, T stage, AUC ROC: 0.72). Most importantly, the addition of CellSearch^®^ count (threshold: 4CTCs.7.5 mL) to D’Amico’s risk factors significantly improved the prediction of metastatic extension (AUC ROC: 0.90), suggesting that the CTCs count carried an important and independent value.

Last, in 47 progressive metastatic castration-resistant PC (mCRPC) patients, Cell-Search^®^ was compared to the AdnaTest and to the polymerase chain detection (PCR) detection of five genes involved in the development and function of the prostatic epithelium (KLK2, PSA, HOXB13, GRHL2, FOXA1) (LE 2c) [16]. The AdnaTest (*p* = 0.027) and PCR (*p* = 0.001) results significantly differed from the CellSearch^®^ count. On Kaplan-Meier curves, detections of CTC by AdnaTest or PCR were superior in terms of survival prediction than those of CellSearch^®^ (≥5 cells/7.5 mL).

CTC in metastatic PC (46 studies, 4322 patients, Appendix A).

In 2009, deBono [73] reported in 276 mCRPC patients that the CTC count assessed by CellSearch^®^ before a new line of chemotherapy was a better predictor of overall survival than any PSA-derived descriptor (LE 3b). Since this seminal report and the Food and Drug Administration (FDA) approval of the assay, CTC count was extensively used in the metastatic setting to predict survival rate and to research actionable characteristics.

In all studies on CRPC but two exploratory reports (LE 4) on biomarkers [21,37], a higher count of CTCs informed decreased the overall survival (OS) rate. This was observed in all types of treatment; from research protocols [11,44,46] to androgen receptor signaling inhibitors, such as abiraterone [32] (LE 4) or chemotherapy with docetaxel [34] (LE 4). Even in the dire situation of CRPC, patients who failed docetaxel, enzalutamide or abiraterone and switched to cabazitaxel, the information gained on survival was substantial (median OS 6.9 and 22.3 months in ≥5CTCs and <5CTCs//7.5 mL, respectively) [13] (LE 2b). A similar correlation was observed for progression free survival (PFS) in CRPC patients switching to abiraterone (LE 2b), enzalutamide (LE 4) [18,36] or cabazitaxel [17] (LE 4). Others addressed the dynamics of CTC counts under treatment, confirming that increasing numbers were predictive of decreased survival [27,36,46] (LE 4), while declining numbers predicted good survival [28,46] (LE 4).

Taken together, all reports on mCRPC consistently highlighted the clinical value of CTC counts at baseline and of CTC dynamics under chemotherapy or androgen receptor signaling inhibitors.

Besides informing survival through their numeration, capturing CTCs also afforded crucial insights on their differentiation.

Antonarakis researched CTCs in 202 mCRPC patients starting abiraterone and enzalutamide and analyzed the respective proportion of full length and splice variant-7 (AR-V7) androgen receptor transcripts. CTC and AR-V7 status impacted PSA progression-free survival, PFS and OS with incrementally poorer figures in CTC negative, CTC positive AR-V7 negative and CTC positive AR-V7 positive patients [8] (LE 4). The PROPHECY study (LE 1b) confirmed in patients under AR pathway inhibitors that pretreatment CTC AR-V7 status was correlated to PFS and OS with poorer response for CTC AR-V7 positive patients who still showed at progression a similar response to taxane chemotherapy [9]. It confirmed the pivotal report by Scher where, after adjusting for clinical measures, mCRPC patients harboring pretherapy AR-V7 positive CTCs experienced better OS with taxanes than with AR signaling inhibitors [40] (LE 2b). Whether the AR-V7 status of CTCs is today strong enough to inform clinical decision is still debated [13] (LE 2b), although there is a tendency in that direction [41] (LE 4). One major limitation to its introduction into the clinical routine is the lack of standardized methods to evaluate and report the AR-V7 status of CTCs, as they varied in the literature from immunofluorescence [10,13,39,40] to mRNA transcript measurements [8,45] (LE 1b). Others reported that patients with CTCs of neuroendocrine differentiation were less likely to respond to AR pathway inhibitors [36] (LE 4).

CTC in localized PC (15 studies, 1450 patients, Appendix A).

### 3.3. Pretreatment CTCs Detection

Not all patients with localized prostate cancer were detected with CTCs [55,57,58,63]. The detection rate varied with the characteristics of the population and the technique used, with figures ranging from 73% with an enrichment-free digital pathology of nucleated cells method [64] (LE 2b) to 50% using a microfluidic ratchet system [58] (LE 4) and 7.5% to 11.2% with the FDA-approved Cell Search system [57,66] (LE 4).

Intriguingly, comparing the detection of cells of epithelial lineage (EpCAM+) in healthy controls and localized prostate cancer patients revealed the presence of CTCs in a minority of controls, accounting for the poor accuracy and sensitivity figures of the technique (53.2% and 40.0%, respectively) [63] (LE 2b). Similar caution in cancer detection was previously voiced in the pioneering paper by Davis [55] (LE 2b), who showed that a comparable minority of cancer patients (21%) and men with elevated PSA, but no tumor detected on extended prostate biopsy (20%) were detected with CTCs by Cell-Search. Of note, as shown in a short series of brachytherapy patients, small traumatisms to the parenchyma may induce in a minority the mobilization of epithelial cells into the blood stream [60] (LE 4), a fact of unknown clinical significance that was also reported at the time of transrectal ultrasound biopsies [74].

Regarding the pathological stage, no correlation between baseline CTC counts [[55],[56]，[57],[58]] was observed in radical prostatectomy cohorts, although highlighting features associated with epithelial–mesenchymal transition in the CTCs was observed in one series associated with extracapsular extension [67] (LE 4).

The hypothesis that in localized PC, biological characteristics were as important as the simple numeration of CTCs was later supported by the observation that the lack of Cluster of Differentiantion 82 protein (CD82) expression on CTCs was associated with poor survival, compared to CTC negative or CTC positive CD82 positive patients who shared the same long-term survival profiles [61] (LE 4). This intriguing result emphasized the value of going beyond simple detection or numeration to get insight into the molecular landscape of the CTC compartment. Here, the presence of CD82, a tumor suppressor gene involved in cell adhesion to protein matrix, showed similar good prognosis as the absence of CTCs. In the same line, the presence of CTCs with high androgen receptor expression was associated with B-cell receptor (BCR) and metastatic progression in a small series of high-risk localized prostate cancer patients undergoing radical prostatectomy [64] (LE 2b).

### 3.4. Post Treatment CTCs Evaluation

To our knowledge, only one study tested CTCs after surgery with the objective of identifying those at high risk of recurrence [62] (LE 2b). Blood samples and bone marrow biopsies were harvested one month after surgery and processed with standard immunochemistry techniques to research “minimal residual disease” in 321 localized prostate cancer patients. Intriguingly, cancer cells in the blood stream or in the bone marrow carried independent information and were potent enough to drive a predictive model that accurately predicted long-term PSA-free survival, independent of the classical predictors of tumor differentiation, PSA, pT stage or resection margins.

Another report, which detailed the dynamics of CTCs numeration by Cell Search during the course of radiotherapy and adjuvant hormone deprivation, failed to highlight any relationships with known clinical predictors or recurrence in 65 patients followed for a median period of 55 months [66] (LE 4).

In conclusion, radical prostatectomy and radiotherapy series confirmed that the presence of CTCs in the blood stream was an early event in the natural history of prostate cancer in some patients. While the simple CTC count was of little bearing, further characterization in terms of differentiation might highlight those with impaired survival expectations.

CTC in PC screening and early diagnostics PC (3 studies, 1455 patients, Appendix A).

Three studies focused on CTC detection. The largest (1223 patients) and most recent used an inhouse assay based on the immunochemistry detection of PSA and P504S-positive epithelial cells [70]. Detecting at least 1 CTC per 8 mL of blood ex-hibited higher sensitivity (0.97, 95%CI: 0.94–0.98), than the classical predictors of PSA density (0.60 95%CI: 0.54–0.65) and % of free PSA (0.42, 95%CI: 0.34–0.44). More importantly, the reported positive likelihood ratio (4.52, 95%CI 3.9–5.1) and negative likelihood ratio (0.02, 95%CI 0.01–0.03) values of the test suggested that a positive test increased 4-fold the odds of detecting cancer, while a negative test decreased 50-fold the odds of a positive biopsy. Again, 18.3% of patients with no cancers detected on biopsies showed CTCs, accounting for a specificity of 0.79. In terms of biopsy strategy, the authors suggested that introducing CTC detection could reduce by 40% the number of biopsies at the cost of missing 3% of clinically significant ISUP 2–3 cancers.

Another study researched those cells of the epithelial lineage before random biopsies using pancytokeratin antibodies and characterized their androgen receptor expression [72]. The expression of the androgen receptor and of the epithelial growth factor receptor was also detailed in the corresponding prostatic tissue. Testing for CTCs was disappointing, with more patients detected with CTCs in those with negative biopsies than positive biopsies (21.6% and 14.3%, respectively). However, it was noteworthy that most CTCs were androgen receptor negative, in line with the relationship existing between low AR expression and epithelial-mesenchymal phenotype, the first step required for the mobilization of epithelial cells into the bloodstream [75].

In conclusion, the use of CTCs in screening and the selection of patients to recommend for biopsies is still in its infancy. The main study was surprisingly positive given the controversial value of CTCs in known cancer patients. The main limitations to the clinical development of the technique in screening are the lack of standardization of the technique, the volume of the blood sample tested, and the limitations of standard biopsies in the era of the MRI-pathway and image-guided biopsies.

## 4. Discussion

The first and foremost information was that competition between commercial techniques of detection, variations in the definition and characterization of CTCs as well as the profusion of clinical outcomes unfortunately obscured this promising field of research.

Most systems are based on the capture in a blood sample of cells showing some degree of epithelial differentiation, such as the expression of cytokeratin (e.g., Cell-Search^®^ [76,77]) or of epithelial membrane markers (e.g., Cell Collector^®^ EpCAM [78]). Such an approach may be biased by the rarity of CTCs that may be missed in a blood sample of limited volume (median sample volume was 10 mL, range from 1 to 40 mL) as well as by the physical presentation of CTCs that may undergo mesenchymal transition [67] and under expression or abrogation of their epithelial markers or present with platelets, stromal cells or monocytes as cellular aggregates that may cloak them from detection [79].

Many original solutions were tested to respond to these limitations; direct capture into the blood stream (Gilupi^®^, [78]), unrestricted automated analysis of all blood nucleated cells (EpicSciences, [80]) or reverse transcription of mRNA of lysed blood cells (ADnatest [16]) before multiplex PCR, capture based on the physical properties of cancer cells (EPIC Sciences [80], HD-CTC assay [81], ScreenCell [82], ISET [83]) including their flow in microfluidic chips [84]. All methods showed some clinical correlation, attesting to the wide potential for innovation in this field of diagnostic and predictive medicine. We feel it necessary to mention that FDA recently granted breakthrough device designation to TriNetra™, a lab-on-chip device, for the detection of CTCs and CTCs clusters in prostate cancer developed by DATAR Cancer genetics [85]. The technology was previously approved in Europe by the National Institute for Health Care Excellence. In the conducted studies the test has confirmed its capability to detect early-stage cancer with accuracy up to 99% without any false-positives. The test does not differentiate prostate cancer subtypes. However, it can identify an underlying squamous cell carcinoma or a neuroendocrine tumor potentially linked with prostate or another primary organ. Appropriate rule-out investigations would be needed to confirm that.

Irrespective of such ingenuity, only the CellSearch^®^ method was FDA-approved and is considered the current gold standard (26/56 studies, Appendix A). The Hamburg group must be congratulated for conducting a robust comparison of this technique with other methods of detection at all three stages of the natural history of PC [16,56,65].

The logistics of the technique are precise, from blood collection in a dedicated tube (7.5 mL, Cell Save Preservation Tube), transfer at room temperature (15–30 °C) to the laboratory where a proprietary automated preparation system (CELLTRACKS^®^ AUTOPREP^®^) is required for the immunomagnetic selection of circulating cells expressing the epithelial cell adhesion molecule (EpCAM) and their subsequent immunostaining. Lastly, a semiautomated fluorescence microscope (CELLTRACKS ANALYZERII^®^) is ultimately used to obtain a semiquantitative characterization of cancer cells. Even though the technology is FDA-approved and was used extensively in breast cancer, melanoma, colon cancer and prostate cancer [86], it is still restricted to research use only.

The current literature in metastatic PC correlated CTC detection and characterization (neuroendocrine differentiation, AR-V7 expression) to the response to androgen deprivation, progression free survival and overall survival [8,10,40,41]. However, the results were less supportive for localized and locally advanced disease, where most authors failed to confirm robust clinical value, except for CD82 CTC status and survival. So, at the current state, CTC detection showed prognostic value in metastatic PCa and even allowed prediction of response to a particular treatment. It may justify the introduction of CTC testing into common practice for such patients. In contrast, obtained data showed no significance for CTC assessment outside of clinical trials in patients with localized disease or suspicious of PCa.

Besides CTC, other subsets of tumor-related cells may be detected in the blood stream, in particular, circulating tumor stem cells (CSC). While CTCs are thought to be predominantly biomarkers, CSCs have distinctive features such as high chemo-resistance and may be directly related to metastasis formation [87]. CSC inhibition may even be applied for targeted therapy in the future [88]. However, the issue of CSC identification and clinical application was beyond the scope of the present review. We intend to conduct a separate review of this topic.

One limitation of our work is the fact that the included articles are rather heterogeneous in terms of methods and outcomes. However, we intended to provide a comprehensive review of all the possible applications of CTC in PC over the last five years and tried to determine the future direction of this issue.

## 5. Conclusions

CTC count and AR-V7 expression showed themselves to be a valuable biomarkers for survival in metastatic PC and response to ARS-inhibitors. CTC diagnostic performance for localized PC or for screening and early detection is not high enough to show additional value over the other biomarkers.

## Figures and Tables

**Figure 1 cancers-14-03802-f001:**
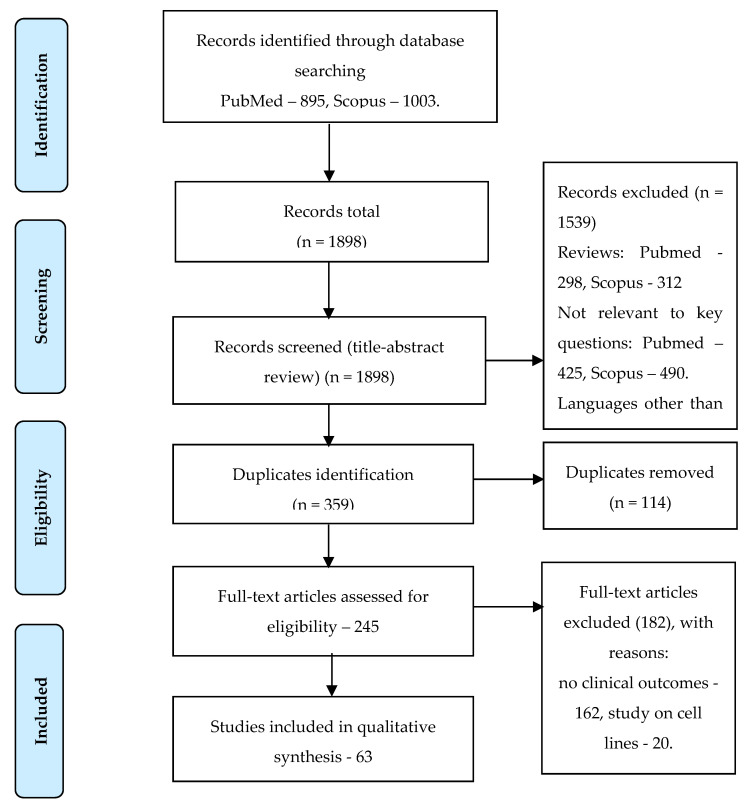
PRISMA statement.

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
