# Peer review of "A Systematic Review of Circulating Tumor Cells Clinical Application in Prostate Cancer Diagnosis"

_cancers, 2022, doi:10.3390/cancers14153802_

Round 1

Reviewer 1 Report

This study was reported the utility of CTC for prostate cancer. The reviewer would like to suggest some critiques as follows.

1.     First, the author should help of a native English speaker prior to submit the manuscript and make more concise this manuscript.

2.     This manuscript has many errors, including appli-cat-ion, sur-vival, spa-tial, etc.

3.     In addition, the method of citation is wrong, including [2],[3], [70]-[72], etc.

4.     On line 22, “clinically important features” is unclear.

5.     On line 22, “ A strong association …  could be observed” is difficult to understand. “CTC concentration was significantly associated with oncological outcomes, including OS and PFS” may be better.

6.     On line 24 and 25, drug names are unnecessary.

Author Response

Reply to the Reviewer 1

This study was reported the utility of CTC for prostate cancer. The reviewer would like to suggest some critiques as follows.

Q1.     First, the author should help of a native English speaker prior to submit the manuscript and make more concise this manuscript.

A1. Thank you for your suggestion. We have consulted a native-speaker and made the manuscript more precise.

Q2.     This manuscript has many errors, including appli-cat-ion, sur-vival, spa-tial, etc.

A2. Thank you. These errors have been corrected.

Q3.     In addition, the method of citation is wrong, including [2],[3], [70]-[72], etc.

A3. Thank you for your comment. We have corrected the method of citation.

Q4.     On line 22, “clinically important features” is unclear.

A4. Thank you, we specified this point adding the phrase “namely survival rates and response to treatment” to the line 22.

Q5.     On line 22, “A strong association …  could be observed” is difficult to understand. “CTC concentration was significantly associated with oncological outcomes, including OS and PFS” may be better.

A5. Thank you for your suggestion. The sentence has been replaced with the following: ” CTC concentration was significantly associated with the overall survival and progression-free survival rate.”

Q6.     On line 24 and 25, drug names are unnecessary.

A6. Thank you. We excluded drug names from the abstract.

Reviewer 2 Report

This systematic review of  circulating tumor cells in prostate cancer diagnosis is a clinically significant overview. However, a key aspect is missing: ie,  'circulating cancer stem cells in prostate cancer' and attempts to target it.  The authors are encouraged to  consult the literature, there are several of this kind of studies: The following articles are just a few of them. Please consult and discuss in the manuscript.

Luo YT, Cheng J, Feng X, He SJ, Wang YW, Huang Q. The viable circulating tumor cells with cancer stem cells feature, where is the way out? J Exp Clin Cancer Res. 2018 Feb 26;37(1):38. doi: 10.1186/s13046-018-0685-7. PMID: 29482576; PMCID: PMC5828305.

Thomas E, Thankan RS, Purushottamachar P, Huang W, Kane MA, Zhang Y, Ambulos N, Weber DJ, Njar VCO. Transcriptome profiling reveals that VNPP433-3β, the lead next-generation galeterone analog inhibits prostate cancer stem cells by downregulating epithelial-mesenchymal transition and stem cell markers. Mol Carcinog. 2022 Jul;61(7):643-654. doi: 10.1002/mc.23406. Epub 2022 May 5. PMID: 35512605.

Author Response

Reply to the Reviewer 2

This systematic review of circulating tumor cells in prostate cancer diagnosis is a clinically significant overview. However, a key aspect is missing: ie,  'circulating cancer stem cells in prostate cancer' and attempts to target it.  The authors are encouraged to consult the literature, there are several of this kind of studies: The following articles are just a few of them. Please consult and discuss in the manuscript.

Luo YT, Cheng J, Feng X, He SJ, Wang YW, Huang Q. The viable circulating tumor cells with cancer stem cells feature, where is the way out? J Exp Clin Cancer Res. 2018 Feb 26;37(1):38. doi: 10.1186/s13046-018-0685-7. PMID: 29482576; PMCID: PMC5828305.

Thomas E, Thankan RS, Purushottamachar P, Huang W, Kane MA, Zhang Y, Ambulos N, Weber DJ, Njar VCO. Transcriptome profiling reveals that VNPP433-3β, the lead next-generation galeterone analog inhibits prostate cancer stem cells by downregulating epithelial-mesenchymal transition and stem cell markers. Mol Carcinog. 2022 Jul;61(7):643-654. doi: 10.1002/mc.23406. Epub 2022 May 5. PMID: 35512605.

Thank you for this valuable suggestion. Indeed, circulating cancer stem cells (CSC) is another important marker addressed in a number of studies. However, this issue was beyond the scope of the present review, in current state comprising 64 original studies. We believe, that including all types of circulating cells in the review, we would make the topic too broad. We intend to conduct a separate review on this topic.

According to your suggestion, we mentioned this topic in the discussion. Namely, we added the following paragraph: «Besides CTC, other subsets of tumor-related cells may be detected in the blood stream, in particular, circulating tumor stem cells (CSC). While CTC are thought to be predominantly a biomarker, CSC have distinctive features such as high chemo-resistance and may be directly related to metastasis formation [86]. CSC inhibition may even be applied for targeted therapy in the future [87]. However, the issue of CSC identification and clinical application was beyond the scope of the present review. We intend to conduct a separate review of this topic.»

Round 2

Reviewer 1 Report

The authors revised the paper in accordance with the reviewers’ comments.

Reviewer 2 Report

The authors have significantly improved the manuscript and may be considered for publication.